# Participatory Crossover Analysis to Support Discussions about Investments in Irrigation Water Sources

**Melle J. Nikkels** [1,2,3,*] **, Joseph H. A. Guillaume** [4,5] **, Peat Leith** [2] **, Neville J. Mendham** [2] **, Pieter R. van Oel** [1] **, Petra J. G. J. Hellegers** [1] **and Holger Meinke** [2,6]

1 Water Resources Management (WRM) Group, Wageningen University, P.O. Box 47, 6700 AA Wageningen, The Netherlands
2 Tasmanian Institute of Agriculture, University of Tasmania, Private Bag 98, Hobart TAS 7001, Australia
3 Aequator Groen & Ruimte, P.O. Box 1171, 3840 BD Harderwijk, The Netherlands
4 Water & Development Research Group, Aalto University, P.O. Box 15200, FIN-00076 Aalto, Finland
5 Fenner School of Environment and Society, Australian National University, Building 141, Linnaeus Way, ACT, Canberra 2601, Australia
6 Centre for Crop Systems Analysis, Wageningen University, P.O. Box 430, 6700 AK Wageningen, The Netherlands
* Correspondence: melle.nikkels@wur.nl; Tel.: +31-620-264-171

**Abstract:** Regional long-term water management plans depend increasingly on investments by local water users such as farmers. However, local circumstances and individual situations vary and investment decisions are made under uncertainty. Water users may therefore perceive the costs and benefits very differently, leading to non-uniform investment decisions. This variation can be explored using crossover points. A crossover point represents conditions in which a decision maker assigns equal preference to competing alternatives. This paper presents, applies, and evaluates a framework extending the use of the concept of crossover points to a participatory process in a group setting. We applied the framework in a case study in the Coal River Valley of Tasmania, Australia. Here, farmers can choose from multiple water sources. In this case, the focus on crossover points encouraged participants to engage in candid discussions exploring the personal lines of reasoning underlying their preferences. Participants learned from others' inputs, and group discussions elicited information and insights considered valuable for both the participants and for outsiders on the factors that influence preferences. We conclude that the approach has a high potential to facilitate learning in groups and to support planning.

**Keywords:** participatory crossover analysis; discussion support framework; personal preference; investment decisions; irrigation water

## 1. Introduction

Uncertainty and complexity, related to changing and variable climatic and economic conditions, create an imperative for strategic and adaptive decision-making on strategies to secure irrigation water availability [1–3]. To enhance adaptive capacity, long-term regional water management plans depend increasingly on investments by local water users [4]. When on-farm investments can substantially influence regional water availability, regional water management organisations need a good understanding of how and when decisions are made to invest in water. If multiple irrigation water sources are available, farmers may display a clear personal preference when comparing alternatives. A personal preference is determined by the sum of an individual's reasoning regarding

options. Reasoning and preferences on irrigation options may vary for many reasons: heterogeneity of local circumstances and situations; real and perceived uncertainties; perceptions of the value of water for irrigation; and tacit knowledge (tacit as in [5]). Generically speaking, personal preferences may differ depending on (1) the set of factors considered, (2) how the factors are understood and integrated into reasoning, and (3) the value that individuals attach to each factor.

Whether implicit or explicit, farmers base their investment decisions on individualised reasoning [6]. Assuming that a group of farmers will uniformly invest if a model indicates a venture to be "worthwhile" might therefore be inaccurate. This suggests the need to better understand the personal reasoning process that underlies decisions on water needs and preferences among sources. Such insight could be particularly valuable to other water users, alongside irrigation scheme designers and water managers.

Crossover points can be used to compare personal preferences and analyse the reasoning underlying them. A crossover point indicates the conditions under which an individual equally favours two alternatives. Analysis of crossover points, expressed as points of indifference, focuses on two key questions: (1) Under what conditions does one alternative out-favour another? (2) What drives personal preference? Crossover analysis is a broadly applicable concept rather than a specific evaluation method [7]. It has been applied for a wide range of purposes:

- to assess the economic feasibility of crop production under uncertainty [8];
- to determine breakeven points in the cost and utilization of managed medical care [9];
- to study points of indifference in pigeons between a small portion of food now versus a delayed but bigger portion [10];
- to determine at what distance from an existing utility line a stand-alone alternative energy system becomes cost-effective compared to a conventional transmission line [11];
- to assess uncertainties in the costs and benefits associated with managed aquifer storage and recovery for improving irrigation water use efficiency at the farm level [12].

The crossover point concept has also been used to explore the sensitivity of modelled outcomes to assumed values for relevant factors in multi-criteria analysis (MCA) [13–15]. Guillaume et al. [15] built on the idea of crossover points to help analysts intuit how crossover points change when adjusting input values in MCA, specifically in regard to irrigation water storage options and the footprints of a vegetarian versus non-vegetarian diet. These authors developed an interactive web interface (link) that visualises the consequences of assumptions on rankings of alternatives. This tool helps analysts to explore crossover points in a learning context.

Discussing crossover points has considerable potential in supporting learning among actors. However, this can best be achieved when the analytical power of crossover analysis is put in the hands of stakeholders. Voinov et al. [16] encouraged the addition of stakeholder experience and expertise in modelling processes. Yet, many existing decision-support applications assume an objective "optimal" outcome based on a decision rule and clearly defined factors that can be captured in a model, such as cost minimisation, in which the cheapest alternative emerges as "best" e.g., [11,12]. Avoiding the assumption of a single "best" option broadens the discussion, as in many cases, "what is best" is far from objective but is, at least in part, a personal preference subject to change over time [17]. It may even be political [18].

This paper contributes to the crossover literature by presenting a framework that extends the use of the concept of crossover points to a participatory setting. The aim is to elicit discussions among water user on investments in irrigation water sources. This is somewhat analogous to what Nelson et al. [19] termed "discussion support" rather than "decision support". The discussion of crossover points in groups is open-ended and subjective, and no single "optimal solution" is pursued. Indeed, personal crossover point indications need not be certain or "right", and no agreement on probabilities is required. This shifts the crossover exercise away from problem solving towards a learning mode, with future uncertainties, personal reasoning, and assumptions at the forefront. The main aim of this new approach,

termed participatory crossover analysis, is to engage participants in a dialogue that explores the personal reasoning process by which preferences are defined. During the discussion, participants receive input from others and contribute their own information and insights regarding qualitative and quantitative aspects of alternative irrigation water sources for the benefit of both the participants and outsiders. Pahl-Wostl [20] considered such informal sharing and integration of knowledge as key for improving water management and governance.

To provide a first, low-stakes test of the framework, we applied it in the Coal River Valley of Tasmania, Australia, where farmers have access to multiple water sources. We begin by presenting the method of participatory crossover analysis and examine its use in the case study. We then explore the implications of the participatory crossover exercise and evaluate the framework. We note that this framework was subsequently applied in a second case study in the nearby South-East Irrigation Scheme district, with a more diverse group of participants and a greater focus on crossover points related to willingness to pay for water [21].

## 2. Materials and Methods

We developed a stepwise framework for participatory crossover analysis that can serve as a checklist for organising a workshop. The framework is formulated in general terms to allow its application in various settings and situations. We tested the framework, with both a practical and a theoretical aim. The practical aim was to facilitate discussions among experienced farmers about irrigation water sources in the Coal River Valley of Tasmania. Specifically, participants discussed how and why their crossover points differed and any changes in their reasoning over time. The theoretical aim was to test whether the framework was applicable (yes or no) and worthwhile (measured by whether participants perceived it as useful). To evaluate the theoretical component, a two-step evaluation process was developed.

### 2.1. Framework for Participatory Crossover Analysis

Figure 1 presents the five-step participatory crossover analysis framework. Step 1 concerns the aim of the exercise. Why will a participatory crossover analysis workshop be useful? What discussion and learning is expected?

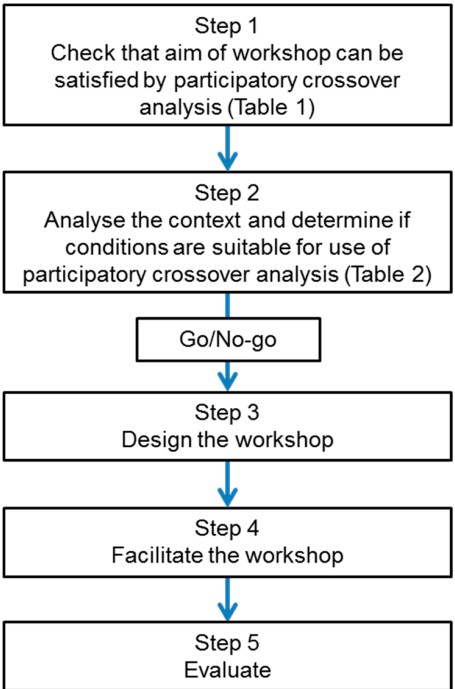

**Figure 1.** Stepwise framework for participatory crossover analysis.

Table 1 lists the aims that participatory crossover analysis can satisfy. Proceed to step 2 only if the aims are clear and suitable.

**Table 1.** Participatory crossover analysis is considered suitable to achieve one or more of the following aims.

| Aims | |
|---|---|
| **Elicit Personal Reasoning** | Participants will be encouraged to share the factors they consider in decision-making, what those factors mean to them, how they integrate them, and the value of each. |
| **Improve Understanding of where Differences in Preferences Come from** | Participants will be given opportunities to reflect on their own personal reasoning and compare it with others, helping them to learn why preferences differ. |
| **Explore Robustness of Personal Preferences** | Participants will learn about the conditions under which preferences change, gaining a sense of their robustness. This encourages them to think about the likelihood that such conditions will occur. |
| **Provide Inputs for Regional Planning Affected by Individual Decision Making** | Sharing decision rules and preferences and providing background information for planning will help participants make or better understand investment decisions. |

Step 2 is to analyse the situation and context of the foreseen participatory crossover analysis workshop. This may involve interviews with proposed participants and should lead to a preliminary identification of alternatives, existing personal preferences, and the factors that influence personal preferences. Table 2 lists conditions required for participatory crossover analysis to succeed. Proceed to step 3 only if all the conditions in Table 2 are satisfied. This may require taking actions that establish suitable conditions, as was done in a follow-up case study [21].

**Table 2.** Participatory crossover analysis is considered suitable only if ALL of the following conditions are met.

| Conditions | |
|---|---|
| **Preferences Are Subjective** | In participatory crossover analysis, there is no objective optimum. Uncertainty is recognized in the assessment of alternatives, and reasoning is understood to be at least partly individual. In other words, what is "best" for me might not be "best" for you. To decide what is "best", we each have our own personal decision rules based on explicit and tacit knowledge. If this condition is not met, a more structured approach could be used (see, e.g., [15]). |
| **At Least Two Discrete Alternatives to Compare** | Participatory crossover analysis requires at least two discrete alternatives to compare, based on one or more factors, which may be uncertain or incomplete. Alternatives may be, for example, whether or not to invest or to adopt an innovation. The crossover concept does not easily translate to continuous decisions, such as how much to invest. |
| **A Dialogue Situation** | Participatory crossover analysis requires an opportunity for a dialogue, for example, a group discussion, in which participants experienced with the alternatives are willing and able to share their reasoning, with minimal reason to withhold information. Participants need to be open to reflection. They must be able to conceptualize the comparison of the alternatives, to express and explore the explanations underlying their personal preferences. |
| **A Facilitator Present** | Participatory crossover analysis requires a facilitator who can handle the range of experience and expertise among participants. The facilitator maintains a safe environment for the participants to share and manages the process in such a way as to "deepen" the dialogue. |

Step 3 is to design the workshop. Design affects both the workshop process and the content of the discussions [22,23]. Think about who will participate. What is their role? Where will the workshop take place? How long should the workshop last? What visual aids might benefit the discussions? The answers to these questions will help determine how the concept of a crossover point should be introduced and what crossover points will be discussed.

Step 4 is to facilitate the workshop. The main role of the facilitator is to quickly pivot from the identification of a preference to the underlying reasoning, aiming to increase the depth of the dialogue, drawing on participants' expertise and experience. Participants are encouraged to identify and expand on influential factors and how these affect their personal preferences. The facilitator informally guides discussions among the participants while looking for (1) differences within the group and the origins of such differences (reasoning), and (2) consensus within the group on factors, reasoning, and thresholds.

Step 5, the last step, is an evaluation process. The aims of the workshop are central herein. An evaluation can be a short recapitulation of the topics addressed and insights gained during the workshop, it can seek information from the participants on the perceived usefulness of the exercise, or the process itself can be evaluated. Additionally, an evaluation can aim to capture tangible outcomes, for example, the impact of the workshop in future decision-making.

*2.2. Testing the Framework*

2.2.1. Case Study Area

We applied the framework to a case study in the Coal River Valley of Tasmania, Australia. The valley is a prime agricultural area in south-east Tasmania (Figure 2). Coal River Valley presented a situation that seemed to meet the first two conditions for participatory crossover analyses; that is, farmers' preferences regarding irrigation water sources were subjective, and there were several alternative water sources that could be compared (see Table 2).

The third condition, a dialogue situation, seemed to be present as well. In 1967, after devastating bush fires, the Coal River Products Association was established to improve cohesion among farmers. It can be seen as a community of practice, as defined by Wenger [24]. The association has been very successful. It was significant in encouraging farmers to try new crops and in building public and political support for irrigation schemes. The elected members of the association's executive committee represent the range of farm enterprises in the valley. All the members knew one another and had a history of knowledge sharing at monthly meetings addressing a range of topics. This gave us sufficient confidence that a dialogue situation could be created during a workshop with members of the executive committee as participants.

Farmers in the valley have gained experience with irrigation since the construction of the Craigbourne Dam in 1986. Since then, the valley has changed more than anyone expected. The direct and indirect benefits of irrigation water were initially hugely underestimated, and farmers have developed their enterprises and intensified and increased their demand for labour [25]. Water demand in the Coal River Valley has thus been on the rise since 1986, leading to the development of other irrigation schemes and recently, to the use of recycled wastewater from neighbouring communities. The valley currently has multiple, very distinct water sources. We selected the oldest, cheapest, and most expensive as relevant to discuss:

- Craigbourne Dam. The Craigbourne Dam is the oldest and first communal source of irrigation water that farmers invested in [26].
- Reuse. Treated wastewater from nearby municipalities is by far the cheapest source of irrigation water. Wastewater from the nearby city of Hobart may offer a way to extend this water source in the future [27].
- SE3. Water from the South-East Stage 3 project provides the most expensive water in the state [28] and commenced operations in October 2015. It could sustainably provide much more irrigation

water than at present, though the development of irrigation schemes depends on investments by both water users and the state [29].

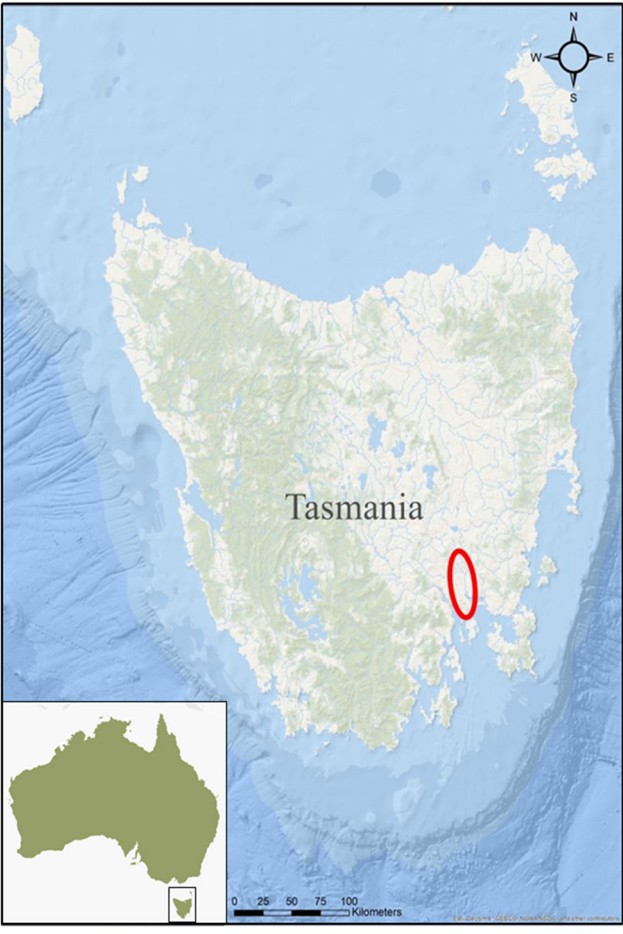

**Figure 2.** Map of Tasmania, with the Coal River Valley in the red circle.

The Coal River Valley is held up as an example of the value of irrigation water for other areas in Tasmania [25]. In this regard, the Tasmanian setting is particularly interesting as a long-term state policy objective is to increase agricultural output through irrigation and innovation [30]. This objective has propelled government initiatives to build new irrigation schemes to facilitate a transformation from dryland cropping to more intensified forms of irrigated agriculture. The approach taken in the design of new irrigation schemes includes a preliminary phase in which sufficient farmers must commit to buying water rights to cover at least 30% of the construction cost of the scheme. The other 70% is covered by the Commonwealth and Tasmanian governments. Commitments at the preliminary stage define the design of the scheme and the diameter, or supply capacity, of the irrigation pipes. As such, regional water availability is influenced by the decisions of water users, though they may be inexperienced in irrigation. It might therefore be beneficial for such farmers—and other stakeholders—to learn from the insights and reasoning of those with experience in making investment decisions on a new irrigation scheme in a comparable valley.

Experienced farmers will likely have garnered skills and information that influence their irrigation water demand and preferred water sources. What would be their preferred source of irrigation water if they had to make an investment decision now? Facilitating a discussion among experienced farmers about water sources might enable farmers to learn from one another. Their insights could provide valuable background information for investment decisions by other farmers, irrigation scheme designers, and water managers.

### 2.2.2. Interviews

In October and November 2016, we conducted in-depth interviews with all farming members of the executive committee of the Coal River Products Association. These 13 persons were also intended to be workshop participants. The interviews lasted 1–2 hours and were geared towards exploring diversity and gaining a better understanding of farming in the Coal River Valley. The interview was set up in two parts. The first part was an accompanied survey to obtain the range of values for initial and operating costs of the various water sources. The second part was more open-ended, asking questions about the context and relevant factors when considering different water sources. We then introduced crossover analysis to each participant and discussed how water sources could best be compared in the workshop setting (Appendix A presents the interview guide).

The interviews were recorded and transcribed. Interview findings were used to check whether all the required conditions were met (Table 2) before proceeding to workshop design.

### 2.2.3. Workshop Design

The workshop was held in late February 2017 in a meeting room at University Farm, where they regularly meet. The 11 participants (2 members could not attend) were seated in a U-form, allowing them to see each other and the facilitator. The workshop was scheduled for an evening and lasted 3 hours. It began with an introduction to the task, followed by two discussion sessions separated by a coffee break, and an evaluation and wrap-up.

The facilitator—the same person who had conducted the interviews—started the workshop by presenting the interview findings, specifically the range of values obtained for the relevant water source characteristics (Table 3). However, the perceptions of these values elicited in the interviews left two key questions unanswered: "Where do these different perceptions come from?" and "How do these differences in perceptions affect personal preferences?" This is what was discussed during the rest of the workshop. To reduce the risk that the discussion would be restricted by actual water accessibility, which varies within the valley, we used a hypothetical case where all three water sources were available and no on-farm infrastructure was yet in place.

The discussion centred on how much one characteristic of the most preferred water source for a type of enterprise had to change before personal preferences shifted to an alternative water source. The facilitator introduced this discussion with a topic question in the form of "How (much) does characteristic X have to change for you to switch from your initial preference to the second-best?" Participants were asked to indicate their initial crossover point via a PowerPoint add-in for polling called TurningPoint [31] and to indicate how confident they were about their crossover point on a personal worksheet. The facilitator then displayed the range and the average of the answers, which were anonymous.

Specifically, the five topic questions were the following:

1. How much does the cost price of water rights for SE3 water have to change before other water sources become relevant for perennial crops? Why?
2. How much does the cost price of the water rights for SE3 water have to change to make it the preferred water source for annual crops? Why?
3. How much value per megalitre (ML, or 1000 m$^3$) do you have to create to still prefer SE3 above alternatives? Why?
4. How much does the reliability of Craigbourne Dam water have to improve to become your preferred water source for perennial crops? Why?
5. What characteristics of reuse water would have to change for it to become your preferred source for perennial crops? Why?

To begin the discussions, the facilitator asked for a volunteer or picked someone, asking them whether the reported change in characteristics, on average, would lead them personally to change their preference. Why or why not? Other participants then expanded on this initial personal reasoning,

adding to the discussion why their own crossover point did or did not differ. The facilitator allowed and even encouraged participants to raise the influence of other characteristics likely to influence the crossover point. After about 15 minutes, or when participants had no more differences to discuss, the facilitator concluded the topic by asking participants to enter their final crossover point in TurningPoint. Again they were asked to record their level of confidence about their crossover point on their worksheet. This time they were also asked to record whether their answer had changed and if so, why.

### 2.2.4. Evaluation of the Workshop

The workshop was recorded with a voice recorder and transcribed verbatim. A note-taker took notes during the process on the usefulness of the discussions in generating transferable content and knowledge. To address the theoretical component, or the process, the note-taker recorded observations on group dynamics, particularly engagement, attitudes, and signs of problems. A twofold evaluation process was employed. First, to provide preliminary feedback on the process, we asked participants to evaluate the workshop and their learning. For this, they filled in an evaluation sheet with both open and multiple choice questions on topics such as their level of comfort in talking honestly about their preferences and personal reasoning, the perceived usefulness of the workshop for themselves and others, and the pace of the workshop (see Appendix B for details). More detailed follow-up came later in the form of telephone interviews two to five weeks after the workshop. These interviews focused on the process and on learning-related outcomes and lasted between 15 and 25 minutes. They were recorded by one of the authors, transcribed verbatim, and analysed thematically. The questions addressed the added value of the crossover concept and the value of the group discussion. Participants were asked what they remembered as particularly useful or interesting and if and how the workshop had changed their thinking and decision-making, as well as the perceived usefulness/value of the discussion to themselves and to others. Appendix C presents the guide for the telephone interviews.

## 3. Results

This section examines the results of the exercise. These are presented first regarding the practical research component, that is, assessments and perceptions of the alternative water sources. Then, results are examined regarding the process, in other words, the theoretical component of the case study.

### *3.1. Practical Component: Case-Specific Results on Water Source Preferences*

#### 3.1.1. Insights from the Interviews

All participants mentioned cost, quality, and reliability as important factors, or "characteristics" as participants called them, in their water source preferences. Table 3 displays the range of values mentioned for the most relevant characteristics. Costs were divided into upfront capital and annual running expenses. The annual component of SE3, which is delivered under pressure (no pumping costs), includes a fixed cost independent of use and a variable cost in relation to the water supplied. The variable cost further depends on the farm's location in the irrigation scheme.

Some participants indicated that sources varied in "manageability", which is related to the ability to trade water with neighbours and flexibility of use (water may be available on demand or be provided as a constant flow over the summer). Some of the relevant characteristics could be defined in different ways, and this might have influenced personal preferences. For example, water quality encompassed an array of parameters and a range of threshold values relevant to suitability for the purpose of an enterprise.

**Table 3.** Water sources and range of values for the most relevant characteristics.

|  |  | **Craigbourne Dam** | **Reuse** | **SE3** |
|---|---|---|---|---|
| **Cost** | **Capital cost per ML (water rights)** | $1000–$2500 | $0 | $2500–$2700 |
|  | **Annual cost per ML at farm gate plus pumping cost to put it in on-farm dams** | $105 plus pumping (up to $150) | $10–$70 plus pumping (up to $150) | $135 fixed + $170–$211 variable |
|  | **Quality** | Variable but often too poor for sensitive crops | Comes with restrictions on applications and crops | Almost drinking-water quality |
|  | **Reliability** | 60–90% | 80–100% | 95% (according to Tasmanian Irrigation) |

Note: ML = megalitre, or 1000 m$^3$.

Participants linked their water source preference and willingness to pay to the crop they grew with the water. In some cases, non-monetary factors were also in play, and these went some way, in certain cases, towards bridging the gap between the cheapest and most expensive sources, possibly making the latter worthwhile. One participant said "The two characteristics I find most important are high reliability and high quality. For that, I pay whatever I need to pay to irrigate my orchard." Another participant said "I will deal with whatever reliability or quality, but I am really focused on cost. Cost is actually all I look at; if it gets higher than I want to pay, I will not grow a crop and will sell my stock." These two quotes represent opposite ends of a spectrum. Based on the recommendations of the participants, we divided the farm enterprises into three types: livestock, annual cropping, and perennial cropping. As Table 4 shows, these enterprises have relative differences in their demands regarding water source characteristics.

**Table 4.** Relative differences of demands for water source characteristics as discussed during the interviews, based on farm enterprise type.

|  | **Livestock** | **Annual Cropping** | **Perennial Cropping** |
|---|---|---|---|
| **Cost (willingness to pay)** | Low | Middle | High |
| **Quality demand** | Low | Middle | High |
| **Reliability demand** | Low | Middle | High |
| **Manageability** | High | Middle | Low |

We also learned that the valuations assigned to characteristics of both water sources and enterprises were subject to change. Indeed, over the years, most participants' willingness to pay for water had evolved. For example, one participant stated: "I remember when water from the Craigbourne cost $15/ML and it went to $20/ML, and we all thought it was too dear. Sometimes you have got to pinch yourself and realise that I'm about to spend $250,000 just to get access to 50 ML of water. If someone would have told me this 10 years ago, I would have thought he was living in fairyland, but perceptions change. If I tell other growers about the reality of irrigation water, they often don't believe me. However, you really need a crop that generates the value that covers the costs."

### 3.1.2. Insights from the Workshop

Our insights from the workshop are focused on the discussion rather than the specific values assigned to the crossover points or their changes. Nonetheless, Appendix D provides an indication of the crossover points. Participants' reasoning is fundamental in determining the crossover points and therefore likely to be more transferrable and relevant to other farmers, water managers, and policymakers than the crossover points themselves. Crossover points, and even changes in crossover points or confidence levels before and after the discussions, may simply be an artefact of the facilitation process (e.g., providing a better understanding of the question). These results are clearly

subject to change, case-study dependent, and by no means representative. There is also a risk that crossover points may be misinterpreted when lacking context.

Our reporting of the discussion focuses on reasoning and insights, with a summarizing sentence at the beginning of every paragraph accompanied by a reflection on the aims in Table 1 at the end of every paragraph.

What is Water Worth?

The first three questions of the workshop focused on willingness to pay for water. Participants discovered that within the group, there were distinct ways of accounting for the various components making up the total cost of water. These contributed to very different views on investments in water rights. The factors considered, the way these factors were brought together, and the assumed cost of the different factors turned out to be subjects of personal perception. Some reported seeing water as a capital cost and spread it over a period of least 10 years. Others just considered the interest rate of their loan to procure water, which would lead to a higher willingness to pay, compared to participants who integrated the cost of water rights into their yearly budget, similar to the purchase of an irrigator or a tractor. Some thought that water would increase in value, while others disagreed. Some expected interest rates to go up in the future, making water more costly if you had to borrow from the bank to finance it. Participants also disagreed on whether a bank would lend money to buy water or not, and about whether buying water is equivalent to buying more land. These different views suggest the usefulness of following up the workshop with a more quantitative study to provide information or advice about strategies to integrate the cost of water rights into a yearly budget.

After several minutes' discussion about the minimum value that needs to be generated per megalitre to still prefer SE3 over alternatives, one participant came up with a rule of thumb. He reasoned, "For me, it would be $6000/ML. I base that on $300 annual cost and 10% of the cost of the water rights, another $300, so $600/ML. I use the rule of thumb that the cost of water should be around 10% of the budget to grow a crop. If you grow fruit, I reckon that if you need more than 10% for your water, you go backwards because you have a lot of other expenses that come in as well; wages are huge costs for me, investment in capital, fertilizer, and marketing." This very explicit line of reasoning began with a discussion on the robustness of preferences, which unfolded into exchanges about this personal rule of thumb. Some participants agreed that although they had not considered the rule before, the 10% was a good figure to aim for. Others reasoned that this figure might be applicable to fruit trees but not annual crops, as water is just one of the many costs involved in growing a high-value crop such as fruit trees. For most annual crops, the percentage spent on water could be greater as there are fewer other inputs. Both the average value of crossover points and the level of confidence (how confident participants were about their crossover point) increased during the discussions. By explaining and exploring the specific rule of thumb, participants gained a better understanding about where differences in willingness to pay for water came from.

There was strong consensus in the group about minimum value generation. Based on their experience and the scale of cropping in the valley, participants agreed that it was impossible to make a profit from either livestock or traditional annual crops (e.g., cereals) using SE3 water. Use of this water source would thus involve a change of enterprise to a high-value crop, preferably "with a contract in your pocket" before investing in water. They did note that the situation might be different for larger farms, as they knew of farmers growing annuals with high-value water in nearby valleys. The finding, based on end-user experiences, that investment in high-value water would require a change in enterprise and everything that comes with such a change, are very relevant for other farmers, irrigation scheme designers, and water managers.

Where does Reliability Come in?

There seemed to be consensus among the participants about the minimum reliability needed for perennial crops: irrigation water bought for use on perennials needed to be at least 95% reliable.

For some, preferences were very robust: Craigbourne Dam water would never be suitable because the quality and the reliability of Craigbourne water was not good enough. The crossover points on reliability and the associated confidence levels stayed the same during the discussions. However, there was much debate about the meaning of reliability and how scheme management affects reliability. One participant said that if there was a guaranteed minimum supply to at least protect your trees from dying, there would be a crossover point somewhere. Another argued that if water was cheap enough, you could buy water rights to have "up your sleeve" if your main source was restricted. Others pointed out management benefits of Craigbourne Dam compared to SE3: (1) "The reality is that the delivery process makes a big difference. When there is not enough water for everyone, water trading kicks in. We learned in the last 20 years that during a drought, some people end up buying water and other people sell, probably making more money than they would have if they applied it to their low-value crop. Craigbourne allows you to buy the yearly water rights from others that do not need it as much." (2) "Craigbourne is a public dam that is holding the water for you. If you buy SE3, you still need a big farm dam, so you are duplicating what is already been done for you." (3) "SE3 gives you water during 180 days a year while you can order Craigbourne water in a large volume delivered over a short time." (4) "Craigbourne actually pays for evaporation while with SE3, you pay for it yourself."

Differences in experiences and in the practical meaning of reliability influenced how participants factored this characteristic into personal decision rules. A preference for other sources seemed very robust if high reliability was demanded but could not be guaranteed.

What Restricts Reuse?

When discussing reuse water, participants agreed that restrictions and regulations needed to be reconsidered, as they were currently hampering uptake. However, they did not agree on which characteristics of reuse water would have to change for it to become the preferred source for perennial crops. Various inhibitive factors were mentioned for reuse water: costly regulations on groundwater monitoring, restrictions regarding empty creeks, regulations demanding that fully grown crops be "washed" with non-reuse water before harvest, and differing regulations for the domestic and international market. Some participants thought that restrictions on reuse water were different in mainland Australia.

Allocation of reuse water was another issue raised as this source is allocated in a year-to-year procedure instead of long-term water rights. Such flexibility in the allocation of water might particularly benefit the water provider, as participants said they would rather know their allocation for at least five years, in order to plan ahead. This indicates that there is room to improve the supply management of reuse water, and a better understanding of the costs and restrictions might influence farmers' willingness to pay. However, "optimal" management is influenced by the perspective taken, as what is best for farmers might not be best for water managers.

*3.2. Theoretical Results: The Participatory Crossover Analysis Process*

Participants differed in their ability to provide or expand on explanations for their initial crossover point. Some seemed initially unable to conceptualise their reasoning. When asked about their initial crossover point, they answered something like "that is just what I think." Nonetheless, after others explained their personal reasoning, they found that they actually could react, compare, and define where and why certain arguments did or did not apply to them. Here, the facilitator played a significant role by encouraging participants to explain their "why".

During the course of the workshop, participants began asking each other more and more questions. The coffee break proved important here, as discussions went on, reflecting, explaining, and comparing—sometimes ending in an agreement to disagree. Once participants began asking each other questions, the discussion really benefited from differences in backgrounds and fields of expertise.

The evaluation indicated that the participants felt willing and able to share their reasoning and listen to each other. They felt comfortable talking honestly about preferences and personal reasoning,

and they were confident that others had been honest during the process (Appendix B). Only one participant did not take part in the discussion, explaining during the later telephone interview that they did not feel comfortable talking, but that listening to others had been very interesting and meaningful. The facilitator and workshop design thus succeeded in providing an environment in which participants were at ease and able to contribute and learn.

Though participants agreed that the process as a whole had been interesting, their opinions were more varied on whether it would influence their decisions. Some said they were already committed to a particular water source, and others had been working in irrigated agriculture for so long that such a short workshop seemed of little influence for them. However, most participants did indicate that the workshop would have some influence on their decision-making, as it contributed to the gradual development of their perspective or intuitive understanding of options, values, and alternatives.

## 4. Improvements and Limitations

### 4.1. Group Composition

In the case presented here, participants varied in their backgrounds and fields of expertise within a farming context. Nonetheless, the group can be considered homogenous as all were experienced irrigators with the same goal: optimising water availability on their own farm. Even more, the participants had a history of knowledge sharing, knew each other personally, and trusted the legitimacy of the process. Thus, the presented case should be situated in the lower left quadrant of Figure 3.

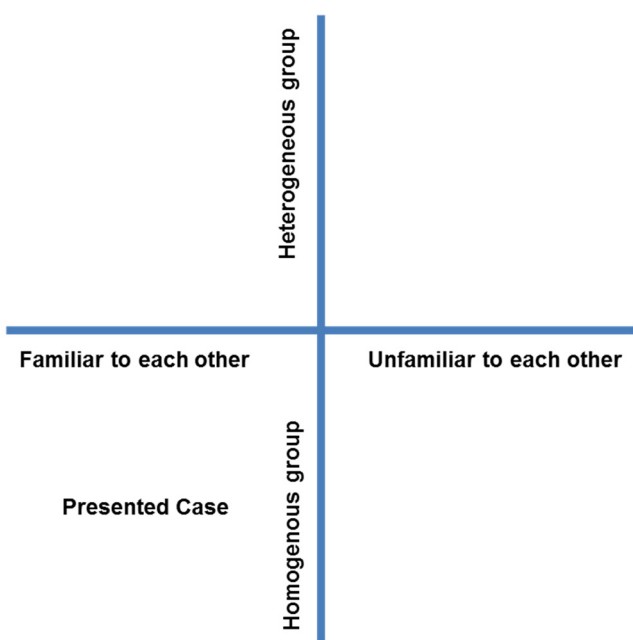

**Figure 3.** The presented case study took place with a homogenous group that knew each other beforehand.

Due to the nature of the Coal River Products Association as a community of practice, participants might have been particularly interested in each other's reasoning beforehand and therefore more open and willing to learn than in cases without a community of practice in place. It would be valuable to test the framework with a more heterogeneous group, in which preferences are based on different backgrounds, perspectives, expertise, and especially, different scopes of involvement in the issue being discussed. An example discussion topic in the Coal River Valley could be "Under what conditions is reuse water the most suitable source to increase water availability in the valley"? Policymakers, water engineers, and farmers could all be involved, providing a heterogeneous group. Such a workshop

could allow stakeholders to "learn together to manage together", which is how Pahl-Wostl et al. [32] define social learning.

There is widespread agreement in the social learning literature that focusing on "how perspectives influence problem definition and preferred outcomes or solutions" is fundamental when managing natural resources like water, e.g., [33–36]. Such a discussion would be situated in the top-left quadrant of Figure 3, as most stakeholders would know each other. In such a setting, the dialogue situation changes in that specific efforts would be needed to ensure that our third condition —a dialogue situation (Table 2)—still holds. In a more heterogeneous group, the legitimacy of the process might be more contested, and participants might have incentives not to share information. Crossover point determination could also be used strategically, for example, in discussing willingness to pay for water in a negotiation setting. In such a setting, the filled-in crossover points become even less relevant. However (qualitative) reasoning might be less prone to strategic use and may still provide a solid basis to help clarify diverse personal preferences. As a method for understanding the reasoning underlying preference outcomes, participatory crossover analysis could become a valuable tool to facilitate social learning, especially in the early phases of participatory processes.

All of our participants indicated that the outcomes of the workshop would be of interest to other farmers, but some mentioned limitations as well. In particular, actively taking part in a discussion was said to provide a greater opportunity for learning than reading about the outcomes of a discussion that others had (see podcast S1). Participants acknowledged that their own learning about how water can be used was a slow and iterative process. Most workshop participants benefited from lessons gained over years. As one participant explained, "When I started farming, I was not irrigating. Then, the Craigbourne scheme came along, basically putting water on my farm for nothing. You just bought your irrigators and started. So, we did not think about the benefits and how much this water is worth. We already had a start. We all were doing something and changed our focus when the water came along. Nowadays, it is very different: you have to buy water and all the infrastructure up front. If Craigbourne would have had the characteristics that SE3 water has today, I wonder if we would have a scheme in the valley. Would we have dug deep in our pockets for it? I wonder if we would be growing crops at all. The focus on perennial crops and with that our water demand, only went up after years of experience."

This statement led to a discussion in which other participants suggested that their "slowly gained" experience could be used to speed up the learning curve for others through crossover discussions. In the Tasmanian setting, it might be interesting to bring experienced irrigators together with dryland farmers from valleys with irrigation potential to exchange knowledge and ideas. This could be facilitated by a crossover approach. Both experienced and inexperienced irrigators could explore and explain their reasoning together, while acknowledging differences in farming context, as in the bottom-right quadrant of Figure 3. This approach would be suitable only for groups in which participants are willing and able to share, explain, and listen.

## 4.2. Workshop Structure

A workshop structured on an explicit model of costs and benefits would have had other objectives and hence produced completely different results. This would likely have led to a dissimilar learning experience for the participants. When asked for recommendations to improve the workshop, some participants did suggest providing a cost model calculating at what point it is "worth it" to invest in water, like the crossover model of Guillaume et al. [15]. Such tools exist in many forms and are being applied, for example, by agronomic consultants to assist individuals and groups in making investment decisions. Although we acknowledge the value of quantitative approaches, our qualitative approach pursued different aims (Table 1).

The added value of the approach presented here is in encouraging participants to challenge established beliefs and to be open and discuss their considerations candidly with each other. It was not aimed at objectively determining the crossover point where a farmer would turn a profit; it was

about learning what a range of farmers considered when faced with an investment decision and where differences in preference came from. Thus, the most appropriate approach would be highly dependent on the aims. Combined approaches could also be applied as combinations can complement each other, see e.g., [37]. A more cost-oriented crossover discussion could be used as a follow-up, especially as workshop participants were found to have distinct strategies for integrating the costs of water in their yearly budgets. Such a workshop could make use of an interface that visualises the consequences of assumptions on cost and benefits on the ranking of alternatives (see [15]).

Participants widely agreed that some of the framing questions in the workshop were confusing. This was in part because some were ambiguous, but also because the idea of discussing a crossover point based on a single variable was initially confusing. During the workshop, the facilitator made clear that the single variable merely defined the angle of the discussion, without precluding other variables from being mentioned. Then, the discussion moved quickly from the observation that selecting preferred options depends on many variables to articulating those variables and, over the course of the workshop, to interrogating each other's thinking and analyses as to why some variables were more important than others. This indicates a need for the facilitator to better explain the angle of the discussions at the start of the workshop.

### 4.3. Capturing Usefulness to Participants

Part of the evaluation focused on the workshop's perceived usefulness to participants. Eight participants indicated in the evaluation that the crossover framework provided a valuable way to support group discussion, or, as one of the participants stated, "I liked discussing irrigation water sources this new way." Nine participants said that the process had been valuable in influencing their thinking about complex water investments. During the phone interviews, most said that they would recommend the workshop to other farmers and agreed that the content of their discussion would be interesting for others. An avenue for future research would be to seek improved means to capture different forms of usefulness and outcomes of workshops in similar complex decision-making contexts.

Participation does not necessarily mean that learning is occurring [38], and evaluating the outcomes of participatory workshops aimed at learning is widely recognised as challenging [39–41]. Participants might find it difficult to indicate that they "learned something", e.g., [42], and might find it even harder to make explicit "what" they learned and how it will influence future investment decisions. In our evaluation, we therefore asked the participants whether the workshop was useful enough for them to recommend it to others or even to participate in another one. If the answer was yes, the workshop was considered likely to have produced new learning or insights. Our evaluation confirmed this, though participants could not directly link their learning in the workshop to specific decisions.

Reflecting on their personal reasoning and learning from and with others to understand why crossover points differed turned out to be both relevant and useful. This learning is, in our case, decoupled from decision making. Decoupling learning from decision making allows participants to bridge divides, moves the discourse away from strategic calculative reasoning, and improves dialogue conditions [42,43].

Participatory crossover analysis deliberately avoids trying to simplify the context and come to a decision that is "best". However, it still asks participants to examine and verbalise their decision-making process. Consciously comparing between alternatives can lead people to focus on an incomplete set of attributes [44], and having to verbalise one's reasoning can produce even larger biases [45]. Besides, focusing on computable factors may be insufficient when trying to solve complex problems [46]. Research in social psychology clearly demonstrates that the more complex a problem is, the less likely it is that conscious thought can contribute much as the subconscious is much better at associating, integrating, elaborating, and weighing in complex situations [47]. On the other hand, intuitive thinking (doing what feels best) is also prone to many biases and conscious thinking (thinking slowly) is often necessary to make the "right" decision in complex situations [48]. Our evaluation results indicate that part of the perceived usefulness lies in the linking of conscious and

intuitive thinking. We therefore suggest that research connecting participatory crossover analysis to the social psychology domain might be particularly fruitful to further improve the framework.

## 5. Concluding Remarks

The participatory crossover analysis framework, as presented, applied, and evaluated in this paper, shows promise in supporting group discussions. We applied the framework in a setting where participants knew each other and shared the common goal of optimising water availability on their own farms. In the case study, different water sources with distinct characteristics were available. Participants engaged in a dialogue exploring the personal reasoning that led to their individual water source preferences. Sharing and integrating local knowledge is said to be key for improving water management and governance [20]. In an informal and explorative setting, participants shared their knowledge and encountered the distinct ways of accounting for the characteristics that determined their water source preferences. The crossover questions focused on the cost, reliability, quality, and manageability of three water sources. Participants discussed (1) how the factors, or "characteristics", under consideration would have to change to switch personal preferences, in other words, for a crossover point to occur; (2) why and how their own crossover points differed from those of other participants; and (3) how participants' reasoning changed over time.

From the start, we were deliberately specific about our aims in organising a participatory crossover analysis (Table 1) and the conditions under which such a discussion could gainfully take place, as the setting was recognised as influencing the process (Table 2). What is required to obtain a productive dialogue situation (condition 3 in Table 2) warrants further exploration to apply the framework in different case study settings.

Our results support the argument that the crossover point concept encourages participants to engage in a dialogue that elicits and explores the personal reasoning underlying preferences and helps explain nonuniformity in investment decisions. During the workshop, participants had the opportunity to share their knowledge and learn from others. A policy implication is that such discussion could provide valuable information and insights on the factors that influence personal preferences. Such information and insights can be of value both to the participants and to others—in this case study, particularly, farmers with the opportunity to become irrigators, as well as water managers and policymakers. Peer-to-peer workshops, such as the one described here, can enrich the knowledge of potential water buyers so that they can make better informed investment decisions. Most workshop participants evaluated the overall process as worthwhile. What they learned, they said, would feed into the gradual development of their thinking and intuitive understanding of irrigation water sources. Moreover, they understood better the reasoning underlying their personal preferences in this regard.

This case study showed the feasibility of applying participatory crossover analysis. Based on the positive evaluations of participants, we believe that the framework merits further development. In particular, we recommend three future research areas when applying the framework in different settings: zooming in on the contexts in which participatory crossover analysis is applicable, assessing the outcomes, and exploring how best to achieve social learning.

**Supplementary Materials:** Podcast S1: Group conversation on water: Why farmers make non-uniform investment decisions. An informal podcast (in Dutch) is available at: https://www.aequator.nl/2019/03/07/phd-podcast-groep sgesprek-over-water-waarom-maken-buren-verschillende-investeringsbeslissingen/.

**Author Contributions:** Conceptualization, M.J.N. and J.H.A.G.; methodology, M.J.N., J.H.A.G., and P.L.; investigation, M.J.N., P.L., and N.J.M.; data curation, M.J.N. and H.M.; writing—original draft preparation, M.J.N., N.J.M., and J.H.A.G.; writing—review and editing, P.R.v.O, P.J.G.J.H, P.L., and H.M; visualization, M.J.N.; supervision, P.R.v.O., P.J.G.J.H., P.L., N.J.M., and H.M.

**Funding:** Joseph Guillaume received financial support from the Academy of Finland-funded WASCO project (grant no. 305471) and the Emil Aaltonen Foundation-funded project "Eat-Less-Water".

**Acknowledgments:** We thank the members of the Coal River Products Association for being so generous with their time and knowledge.

**Conflicts of Interest:** The authors declare no conflict of interest.

**Appendix A. Interview Setup**

*Part 1: Accompanied surveys to obtain the range of values for the initial and operating costs of the various water sources and how they are used by farmers.*

General on property:

- What crops and pastures do you grow?
- On how many hectares or on what area (1 ac = 0.4 ha)?
- How many hectares do you have in total?
- What is the storage capacity of your system? (Farm dam) (ML = 1000 $m^3$ = 100 mm/ha = 250 mm/ac)
- What role does the farm dam play in your water supply system?
- What types of irrigation do you use for your different enterprises?
- How much water do you use for irrigation per year?
- How does that vary over the years (min/max)?
- How valuable is water for your different enterprises?
- How much value do you generate per ML?

Water sources:

- What sources of water are available to you?
- What sources of water do you use? We need to understand why, thus the following questions...
- What are important factors to you when considering different water sources (quality, quantity, security)?
- What are the costs of each source? Let's break these costs down to various components.
- What are the initial costs: water rights, construction costs for infrastructure (including drip lines, pumps and/or irrigators of various sorts)?
- What are the operating costs over the lifespan of the system (How often do you replace parts of the infrastructure? How long will the infrastructure last? What maintenance is required? What is the rate of return on investments? What other costs are involved in getting your water "to the right place at the right time"?
- How did these costs change in the past? How do you think they will change in the future?

Comparing sources

- How reliable are your different sources, and how does this affect your usage of them?
- What are the benefits of each source?
- How did these benefits change in the past? How do you think they will change in the future?
- What are the risks of each source?
- How did these risks change in the past? How do you think they will change in the future?

*Part 2: Semi-structured interviews to understand context and design for a hypothetical farm*

- What is your preferred water source? Why?
- How can you increase water availability on your farm?
- What are relevant water sources that could become available in the near future?
- What farm characteristics define water demand?

Interviewer explained crossover approach and asked for input:
*During the cross-over session, you will discuss the ranking of different water sources, based on the relevant cost and benefit components. We will focus on the questions: "Under what condition would your initial ranking*

*change? How robust are your current preferences? Why do you prefer certain water sources, and how could this be taken into account when designing a new scheme or expanding current sources?*

*For the proposed workshop, we still have scope to change things to make sure it is relevant to you. We are thinking about using a hypothetical farm as a basis for discussion and analysis. We will start from scratch and assume that all or most sources are available. What would be the types of costs of the different sources, and what would be the amount of water needed to irrigate certain crops? Or, if you were the owner of this farm, what sources of water would you invest in and why? What would the hypothetical farm look like and what sources and strategies for water supply would be relevant?*

*Another option would be to discuss water sources by focusing on production. We could say, we generally have three categories of production, those being perennial horticulture (cherries, grapes, etc.), mixed crops (seed crops, poppies, cereals, etc.), and livestock (sheep, lamb, cattle). Let's say we deal with them separately, as they are very different enterprises. So, if you grew cherries or grapes on, let's say, 10 hectares, how much water do you need and what water source would you like best?" (Same for mixed crops and livestock)*

- How do you think that water availability could change in the Coal River Valley in the future? Why do you think that? How will you respond to those changes?

- Where do you see your farm in 20 years' time? How do you think the Coal River valley will develop?

## Appendix B. Group Evaluation Questions and Outcomes

**Table A1.** Results from the evaluation immediately after the workshop.

| Environment | | | | | |
|---|---|---|---|---|---|
| | Strongly disagree | Disagree | Neither agree nor disagree | Agree | Strongly agree |
| I felt comfortable talking honesty about my preferences. | 0 | 1 | 0 | 6 | 4 |
| I believe others in the group were consistently honest throughout the workshop. | 0 | 0 | 1 | 10 | 0 |
| I felt comfortable talking about my reasoning for preferences. | 0 | 0 | 1 | 8 | 2 |
| The workshop facilitation was appropriate for the content and group. | 0 | 0 | 0 | 10 | 1 |

| Workshop | | | | | |
|---|---|---|---|---|---|
| | Strongly disagree | Disagree | Neither agree nor disagree | Agree | Strongly agree |
| If I talk about the workshop to other people it will mostly be positive. | 0 | 0 | 4 | 7 | 0 |
| The outputs of this workshop should be interesting to other audiences. | 0 | 0 | 3 | 7 | 1 |
| The pace of the workshop was: | Variable | Very slow | A bit slow | About right | A bit rushed | Too fast |
| | 2 | 0 | 1 | 8 | 0 | 0 |

| Crossover Points | | | | | |
|---|---|---|---|---|---|
| On average, other people in the group had preferences that were: | I hadn't really ever thought about it | About the same as I expected | Slightly different from what I expected | Very different from what I expected | |
| | 3 | 4 | 4 | 0 | |
| | Strongly disagree | Disagree | Neither agree nor disagree | Agree | Strongly agree |
| The crossover approach has added something to the way I will think about water investment decisions. | 0 | 1 | 3 | 7 | 0 |
| The crossover process helped to inform my thinking about water investment decisions. | 0 | 0 | 2 | 8 | 1 |
| The crossover framework is a valuable way to guide group discussion. | 0 | 0 | 3 | 6 | 2 |

**Appendix C. Guide for Follow-up Evaluation Phone Calls, 3–5 Weeks after the Workshop**

(This first question uses an inductive open-ended approach to elicit stand-out memories and take people back to the event and the discussion going on there.)

(1)　Ok, so the first question is about any general reflections on the workshop and the discussions you had last week. Were there any parts of the discussion that stood out or that you remember as particularly useful or interesting? Are there things that surprised you about the perspectives of other people in the group? (Follow-up: Why was that interesting/useful?)

(Question 2 elicits thinking about the use of the process and outputs for learning.)

(2)　This question is about potential value of the crossover process in meeting its goal of enabling groups to learn and potentially improve decision-making. (The process is understood to extend from the interviews to the workshop and writing up the findings). So, for the following groups, what do you see as the potential value for learning and decision-making:

|  | Potential value (learning and decision-making) |
|---|---|
| For the group of Coal River irrigators (workshop participants) |  |
| Other farmers in Coal River Valley |  |
| Farmers from other valleys who are considering irrigation investments or recently got access to irrigation water |  |
| Policymakers and utilities (e.g., Taswater, Tas Irrigation) |  |

(Question 3 seeks input for future improvements)

(3)　Are there any ways that you think the crossover process could be adapted or improved to make it more useful or achieve its full potential?

○　Focus on important characteristics
○　Other people that you think would have been valuable in the discussion:

|  | Reason |
|---|---|
| Other farmers from valley |  |
| Other farmers from elsewhere |  |
| Politicians |  |
| TI |  |
| TAS Water |  |
| DPIPWE |  |
| Others |  |

○　Who should facilitate these discussions? Was it good to have an independent researcher, or could the facilitator be from TI, DPIPWE, or MAQFRANK?
○　Discussion Support System and modelling?
○　Different presentation formats and tools?

(This fourth question is geared towards impact and robustness, or changes in ways of thinking and deciding.)

(4)　Did the discussion give you a better understanding of or confidence in your water source preferences? If so, can you say what the influence was?

○   Did you get a better understanding of where differences between neighbours in crossover points come from?

○   Did you continue the discussion with others?

■   Did that produce new answers or insights?

■   Would you have filled in other values if you could do it again? If so, for which question and why?

Anything else:

## Appendix D. Crossover Indications

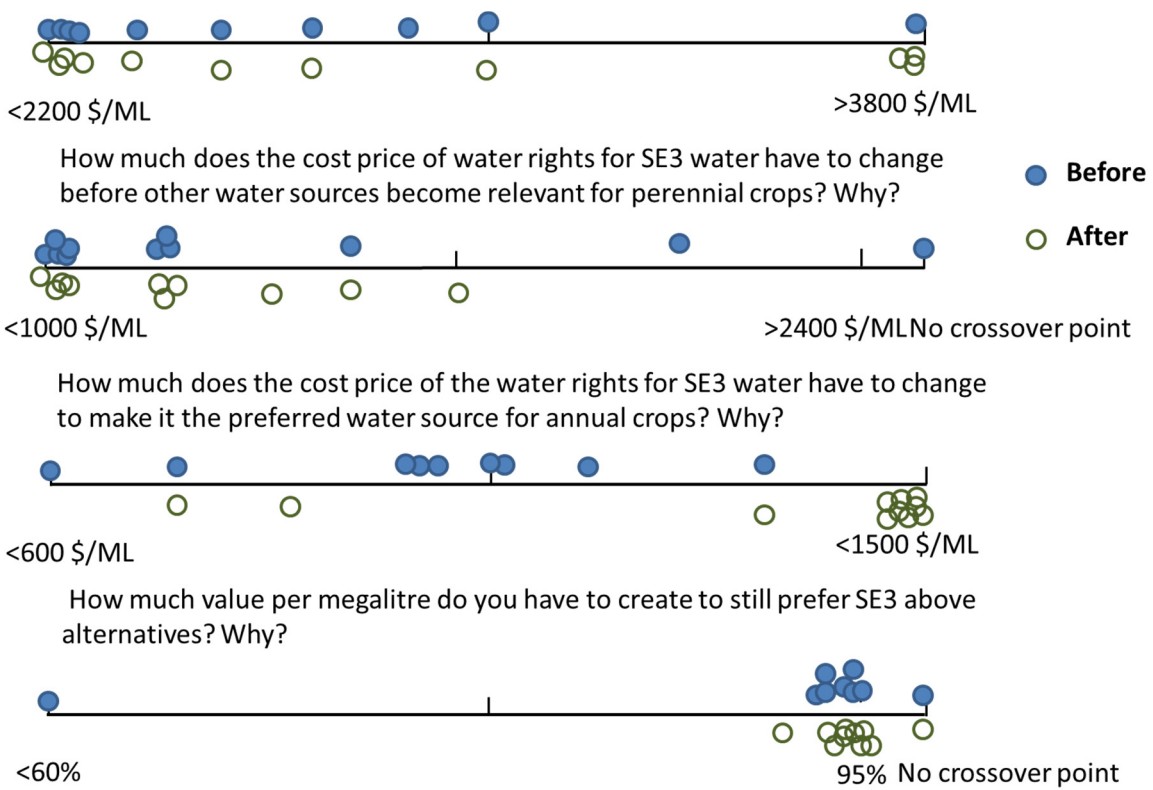

**Figure A1.** Crossover points of the participants with the crossover point before the discussion above the line and the crossover points after the discussion displayed below the line. The primary intent of asking for crossover points is to start as discussion; crossover points can not be understood as stand-alone results or willingness to pay. These crossover points are clearly subject to change, case-study dependent, and by no means representative. The Figure just shows the different perspectives within the group. Changes occurring during the discussion might be due to learning but also simply better understanding the question at hand.

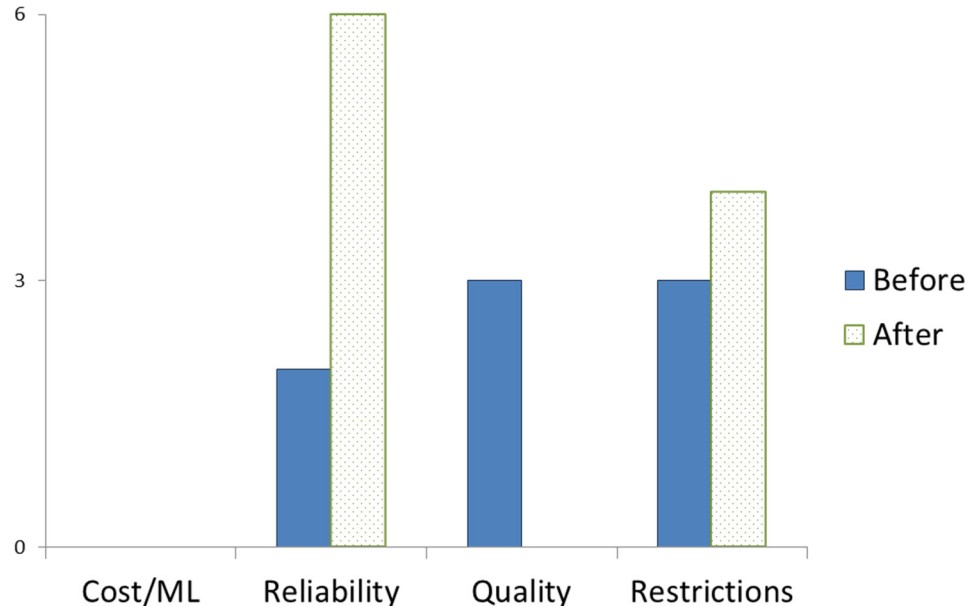

**Figure A2.** Participants' indication of the most important characteristic of reuse use water before and after discussion.

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
