# Peer review of "Participatory Crossover Analysis to Support Discussions about Investments in Irrigation Water Sources"

_water, doi:10.3390/w11071318_

Round 1
Reviewer 1 Report
1. The paper focuses on the methodology proposed by authors and they focus its convenience in a case study where, at least for me, seems to fit too easily. If the idea is to prepare workshops to ease the discussion between agents in an area, what’s the point to prepare the case in an area where agents have being in touch for 50 years? Are the good results of the “experiment” due to the methodology proposed by authors or it is due to the concrete conditions of the case study that eases the discussion even without applying the methodology?
2. In the introduction authors show that the crossover analysis has been used for a wide range of purposes but none of the purposes presented is related to water management… Has this methodology not been applied in any case related to water management?
3. Authors explain that this methodology is not used to achieve a “best option” but, could be useful to achieve “best options”? That is, not trying to find the best one but to find different outcomes that can be considered as best options depending on the characteristics analysed in the crossover analysis? Also, why not trying to complement their qualitative approach with a quantitative one? It could improve the valuation of the methodology for users in the future. If possible, maybe authors can create a basic guide to complement both approaches.
4. The methodology seems well structured and explained but I have some comments, first of all, in general for me it seems that the conditions of table 1 are not too difficult to accomplish… So, it seems that the crossover analysis can be applied for any case, is it true?
5. Authors explain briefly that an evaluation can aim to capture tangible outcomes. Maybe authors should focus more in how it would be useful the use of their methodology to get tangible outcomes that are useful for participants. Not only getting a discussion between participants but also how the discussion can help them to ease their decisions on the topic analysed.
6. Why using only this three water sources for the workshops. I suppose that it’s due to the previous interviews with the agents, but it’s not explained properly in the paper. Maybe it would be interesting for authors to justify why using only this three water sources. Also, why authors classified farmers into three groups (and why only three?), it could be logical but it’s not explained through the text and readers could be interested in knowing what the reasons behind this classification are.
7. About the questions, how authors know that participants are not answering strategically to the questions related to costs? Although it’s understandable that in their case study participants could be more open-minded, could be a problem asking this type of questions in a more heterogeneous group? Is there any way to avoid or reduce the possibility strategic answers?
8. I do not see the purpose of the part about the link with complex decision-making contexts. It seems that it has been including as an extra. I would like to know the justification of this part. That is, the paper is about a new methodology that it’s not linked in any part of the paper with the focus on decision-making contexts and then, at the end, authors drop that idea… On that topic, it should 4.4 instead of 4.3, isn’t it?
9. Concluding remarks should be better explained and I would like authors to explain how it could be improved this methodology with new study cases or different approaches. That is “a future research” using their methodology as a starting point.
10. There are some sentences that seem out of context. What’s the contribution of saying, in the text that, “the main role of the facilitator is to draw focus (…) asking “why”? It seems, for me, totally unnecessary. In addition, it is relevant to explain how the participants were seated in the workshop? Also, what’s the point of explaining the schedule of the workshop? I am not saying it’s not relevant but, if it’s relevant, then it’s not explained on the text.
11. Finally, at line 518 what’s the point of including the comment of only one participant to justify how useful is the approach? It’s the comment of only one participant…
Author Response
Dear Reviewer,
We have carefully revised the manuscript following your constructive comments. We respond to all comments and indicate the corresponding actions taken. The indicated line numbers correspond to the manuscript with track changes.
Please see the attachment.
Yours sincerely,
The Authors

Reviewer 2 Report
The study presents an interesting topic, basically in the fields of Decision Theory. The methods and the results are well written. I added a theoretical aspect and some practical inquiries to improve the article's logical presentation.
In the introduction section (lines 40-49), you describe a process of preferences’ evaluation. Especially lines 48-49 and 51-54 are very similar to the multi-criteria analysis (MCA). A question that arises here to the reader is why you prefer crossover points instead of MCA.
Recent studies have attempted to evaluate different MCA techniques for the same purposes, to find the most suitable one, and apply it to different groups of interest. A reference you could add for example is the following:
Alamanos, A., Mylopoulos, N., Loukas, A., & Gaitanaros, D. (2018). An integrated Multicriteria Analysis tool for evaluating water resources management strategies. Water 2018, 10, 1795; doi:10.3390/w10121795.
I recommend adding a short comment with the advantages of crossover analysis compared with MCA, or the reasons you prefer it. Thus, the readers would feel safer and more justified to follow your framework.
Lines 119-120 and 125: What if not all the conditions are satisfied? How do you evaluate this situation and how do you proceed in this case?
The analysis based on a workshop needs a sample of participants that are already willing to cooperate and learn. Otherwise they wouldn’t participate in such a process. In most areas with water scarcity and complex economic decisions, worldwide, the conditions are not ideal for the enrollment of such approaches.
I suggest adding a comment or a short answer to the following questions arising:
a) Is this objective (e.g. Fig.3), and is the sample representative?
b) What if strong disagreements occur?
c) What if the sample cannot reflect exactly their preferences and reasoning?
d) How can you work in an area where the stakeholders do not have the background (educational, willingness to learn and cooperate) to participate?
e) Is there any significant differences between the political decision-making and planning, and the stakeholders’ preferences? How do you manage such a situation (or a situation where the political will cannot be influenced by the sample’s preferences)?
Author Response

(The authors gave the same response as above.)
